# Factors associated with measles vaccine immunogenicity in children at University Teaching Hospitals, Lusaka, Zambia

Priscilla Nkonde Gardner[1,2]*, Jimmy Hangoma[1,3], Cephas Sialubanje[2,4], Musole Chipoya[2], Lillian Lamba[2], Musaku Mwenechanya[5], Rodgers Chilyabanyama[6], Mpanga Kasonde[2], Davie Simwaba, Muzala Kapina[2], Soo Young[7], Kelvin Mwangilwa[2], Roma Chilengi[2], Isaac Fwemba[1]

**1** University of Zambia, Institute of Distance Education Lusaka, Zambia, **2** Zambia National Public Health Institute, Lusaka, Zambia, **3** School of Health Sciences, Levy Mwanawasa Medical University, Lusaka, Zambia, **4** School of Public Health, Levy Mwanawasa Medical University, Lusaka, Zambia, **5** Department of Child Health, University Teaching Children's Hospital, Lusaka, Zambia, **6** Health Programs, Resolve to Save Lives, Lusaka, Zambia, **7** Health Programs, Churches Health Association of Zambia, Lusaka, Zambia,

* priscillagardner82@gmail.com

## Abstract

Measles poses a significant global public health challenge, particularly in low-resource settings where vaccination coverage is limited. This study examined factors associated with measles vaccine immunogenicity in children aged 2–15 years at the University Teaching Children's Hospital and the HIV Pediatric Centre of Excellence in Lusaka, Zambia. This comparative analytical cross-sectional study was conducted from April to July 2024, enrolling 200 children, including 100 HIV-positive and 100 HIV-negative participants. All children had received at least two doses of a measles-containing vaccine and had no history of measles infection in the past six months. Blood samples were analyzed for measles immunity, while data on age, HIV status, breastfeeding history, and socio-demographic factors were collected. Among HIV-negative children, 75% retained immunity, whereas only 38% of HIV-positive children retained immunity. Multivariate logistic regression showed that children aged 10–15 years were less likely to retain immunity compared to those aged 2–4 years (AOR = 0.270, 95% CI [0.114–0.618], p = 0.002). HIV-positive children had lower odds of retaining immunity compared to HIV-negative children (AOR = 0.290, 95% CI [0.137–0.594], p < 0.001). Breastfed children had higher immunity retention (AOR = 0.336, 95% CI [0.147–0.738], p = 0.007) compared to non-breastfed children. Residing in Lusaka was associated with lower immunity retention (AOR = 0.250, 95% CI [0.066–0.859], p = 0.031). These findings highlight the protective role of breastfeeding and suggest that older and HIV-infected children may benefit from booster doses to sustain measles immunity.

**Data availability statement :** The data is available online deposited in figshare repository and the link to the file 10.6084/m9.figshare.27645726 [38]

**Funding:** The authors received no specific funding for this work.

**Competing interests:** The authors have declared that no competing interests exist.

## Introduction

Measles infections are a significant global public health challenge, particularly in low-resource settings where vaccination coverage and access to healthcare may be limited [1]. Immunization remains the most effective strategy for reducing morbidity and mortality from infectious diseases worldwide. However, despite the availability of effective vaccines, measles continues to cause substantial morbidity and mortality. In 2021, the World Health Organization (WHO) reported over 9 million measles cases and 128,000 measles-related deaths globally, with most fatalities occurring in children under five years of age [2]. Measles also results in severe complications such as pneumonia, encephalitis, and subacute sclerosing panencephalitis (SSPE), a progressive and fatal brain disorder that can develop years after the initial infection [3].

In Zambia, measles remains a persistent challenge despite ongoing immunization efforts. Between 2022 and 2023, the University Teaching Hospital (UTH) recorded over 600 measles cases, resulting in approximately 20 deaths. Notably, the majority of these cases affected children aged 1–10 years, highlighting the vulnerability of this age group and the need for strengthened vaccination strategies and monitoring systems [4].

The immune response to vaccines may vary considerably due to factors such as age, HIV status, and socio-demographic conditions [5,6]. Previous research has shown that despite receiving antiretroviral therapy (ART), HIV-infected children often exhibit weaker immune responses to vaccines, including the measles vaccine, compared to their HIV-negative peers [7]. HIV-infected infants who discontinue ART have been shown to be more likely to lose their measles immunity by age 4.5 years compared to those who remained on ART [8]. According to the 2024 Zambia Demographic and Health Survey (ZDHS), the HIV prevalence among children aged 2–14 was estimated at 1.3%. Given Zambia's population structure, this percentage translates to approximately 80,000 children living with HIV. This significant number underscores the critical need to address HIV in pediatric populations, especially considering the potential for co-infections such as measles, which can further compromise the health of immuno-compromised children. While the ZDHS does not provide specific data comparing measles vaccination rates between HIV-positive and HIV-negative children, it is recognized that children living with HIV may face additional health challenges that could impact vaccination coverage and effectiveness. Factors such as access to healthcare services, socioeconomic status, and potential immunological differences may influence vaccination rates and responses in this population. Further research is needed to assess disparities in vaccination coverage between HIV-positive and HIV-negative children in Zambia. Such studies would inform targeted interventions to ensure equitable access to immunization services for all children, regardless of HIV status.

Age plays a critical role in measles immunity. Studies on measles vaccine effectiveness, such as those by Franconeri et al., have demonstrated that immunity wanes over time, with older children, particularly those aged 10–15 years, showing significantly lower levels of seropositivity compared to younger children. While these studies measure vaccine effectiveness in real-world settings using seropositivity as a

marker, the findings underscore the need for careful monitoring of immunity levels across age groups and consideration of booster doses to sustain population-level immunity [9,10]. A study by Puthanakit in Thailand found that older children had significantly lower rates of measles seropositivity compared to younger children, highlighting the need for age-appropriate vaccination strategies [11]. Additionally, breastfeeding has been associated with enhanced vaccine immunity, as breast milk contains immunological factors that boost early-life immunity [8]. The interplay between these factors age, breastfeeding, and HIV status warrants further investigation to guide tailored vaccination strategies and improve immunization outcomes.

The co-occurrence of HIV and measles in Zambia raises concerns about the effectiveness of current vaccination programs, especially among HIV-infected children. Current evidence suggests that, despite high vaccination coverage and access to ART, HIV-infected individuals may remain vulnerable [8]. This highlights the need for improved and sustained immunization strategies for this population. Supplementary immunization activities (SIAs) have been identified as great opportunities for reducing inequalities in measles vaccine coverage, especially in rural and hard-to-reach areas. However, studies in Zambia have pointed out challenges in implementing SIAs, especially during the COVID-19 pandemic complicating efforts to achieve optimal measles vaccination coverage in the country [12].

The measles prevention initiative in Zambia seeks to reach 95 percent of children each year through the routine immunization program [13]. The country recommends two doses of a measles-containing vaccine (MCV) to be delivered through the routine program at 9 months and 18 months of age as a minimum optimal number of immunizations [14]. However, a study on factors associated with zero-dose vaccination status in Zambia's Southern Province revealed that gaps in vaccine coverage persisted even after mass vaccination campaigns [15]. This gap may present a considerable risk, especially for children living with HIV [16]. A seroprevalence study in Zambia highlighted disparities in immunity levels between HIV-positive and HIV-negative children and adults, emphasizing the need to address systemic issues like cold chain management and the unique requirements of immunocompromised groups [12].

Despite these insights, there remains a gap in understanding the effect of age, HIV status, and socio-demographic factors on measles vaccine immunogenicity in sub-Saharan African settings, including Zambia where many children and adolescents are living with HIV. This study aimed to examine factors associated with measles vaccine immunogenicity in children aged 2–15 years at University Teaching Hospitals in Lusaka, Zambia. Specifically, the study sought to answer the following questions:

1. How does breastfeeding status influence measles vaccine immunity in children?

2. What is the impact of HIV status on measles vaccine immunogenicity?

3. How do Age, sex, educational level and place of residence affect the retention of immunity in this population?

Insight into these factors were important to inform policy and interventions on measles vaccination programs in Zambia.

## Methods

### Study design

This study employed a prospective comparative analytical cross-sectional design to evaluate factors influencing measles vaccine immunity. Data collection was conducted prospectively from the 2nd of April to the 31st of July 2024. The prospective nature of the study allowed for the collection of up-to-date and comprehensive data, ensuring high-quality analysis of immunogenicity determinants.

### Study setting

The study was conducted at the University Teaching Children's Hospital and the Pediatric Centre of Excellence (PCOE) in Lusaka, Zambia, both of which have a high prevalence of measles and HIV. The hospital recorded over 600 measles cases with about 20 deaths between 2022 and 2023 [17].

## Study population and sampling procedure

The study population consisted of children aged 2–15 years who had received at least two doses of the measles vaccine, with participants stratified based on their HIV status. The sample size was calculated using the formula for comparing two proportions, accounting for a 10% non-response rate:

$$n = \frac{2(Z_a + Z\beta)2 \cdot \left(P1(1-P1) + P2(1-P2)\right)}{(P1-P2)2}$$

The measles prevalence rate for the infected group (P1) was 47.2% (0.472) [18], while the uninfected group (P2) had a prevalence rate of 85% (0.85) [18]. For this analysis, 'infected' refers to individuals living with HIV, whereas 'uninfected' refers to individuals without HIV. The study employed a significance level (α\alphaα) of 0.05, corresponding to a 95% confidence level, and a statistical power (1−β) of 80%, ensuring sufficient sensitivity to detect meaningful differences. The Z-score for a two-sided test at α/2 was approximately 1.96, while the Z-score for 80% power was around 0.84. This resulted in a working sample of 98 participants per group, leading to a total of 196 participants, rounded to 200.

## Inclusion and Exclusion Criteria

Participants who were included in this study met the following criteria:

• Children without measles infections in the past 6 months based on past medical records and clinical interviews.

• Children aged between 2 and 15 years

• Available and complete medical records with immunization history and HIV status,

• Documented measles vaccination status with at least two doses of a measles-containing vaccine

• Confirmed HIV infection diagnosed according to national guidelines

• HIV-non-infected children without a history of HIV infection or exposure

• Assent from children and consent from their parents or legal guardians

Children below the age of two years and above 15 years were excluded from the study, as were those with severe immunodeficiency other than HIV, such as primary immunodeficiency disorders.

In addition, children with chronic illnesses or medical conditions that could affect immune response or vaccine immunogenicity, such as cancer, were also excluded. Further, children who had received blood products or immunoglobulins within the past three months, those currently participating in another clinical trial involving vaccines or immunomodulatory agents, and children with incomplete medical records or missing information about measles vaccination status or HIV diagnosis were excluded. Additionally, children whose parents or guardians declined participation, and those unable to provide assent or cooperate with study procedures due to cognitive or developmental limitations were also excluded.

Fig 1 illustrates the participant recruitment process for the study, encompassing screening, exclusions, and final outcomes. Initially, 330 participants were screened for eligibility. Among these, 51 participants declined to provide consent, 29 were excluded due to critical medical conditions, 20 were excluded for being below the age of 2, and 30 were excluded for being over the age of 15.

Following the application of the inclusion and exclusion criteria, a total of 200 participants met the eligibility requirements and were successfully enrolled in the study. Among these enrolled participants, 101 (50.5%) retained immunity, while 99 (49.5%) experienced waned immunity. The flowchart provides a concise and clear visualization of the recruitment process and the resulting distribution of participant outcomes.

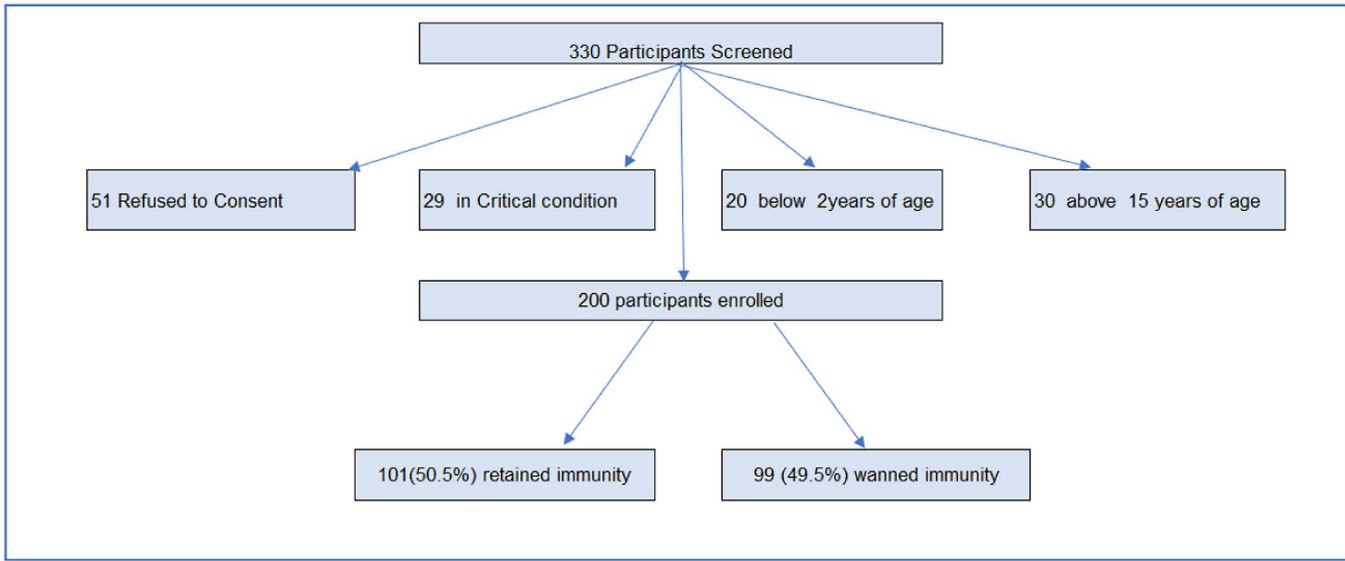

**Fig 1. Participant recruitment algorithm.**

## Data collection

Kobo a digital data collection tool was used for real-time data entry and storage and enabled the research assistants to gather primary data from both sick and healthy participants who visited the University Teaching Hospital (UTH). Phlebotomists and data collectors made up the field team. These data collectors spoke Nyanja fluently, as did the phlebotomists (blood collectors). They worked under the direction of a team leader and received training in research ethics, biosafety, quality assurance, and the proper use of personal protective equipment. Additionally, the phlebotomists received training on the proper techniques for performing and drawing blood. Supervisors and data collectors received training on how to use the tablet to complete the questionnaire. The field teams also had safety handling procedures and infection prevention and control (IPC) tools. Close supervision, data editing and cleaning, and cross-checking of the surveys' completeness were further methods used to regulate the quality of the data. Pre-testing the questionnaire in comparable contexts outside the research area allowed for essential revisions to be made to a few of its items. Blood samples of about 2mls were collected from study participants by phlebotomist and health care workers and using the available hospital courier systems, the collected samples were transported to UTH virology laboratory for testing. The testing was conducted in line with the laboratory protocols for measuring measles-specific antibody titres in Zambia. Measles IgG test kits were procured in consultation with the Zambia National Public Health Reference Laboratory. Laboratory quality control procedures were performed, and the test kits met compliance standards. Measles-specific IgG antibody titers were quantified using Origene quantitative test kits (EA100951). Participants were classified as having retained immunity if their IgG titers were ≥120 mIU/mL, based on the manufacturer's specifications, while titers below this threshold indicated waning immunity. To ensure methodological rigor, the ELISA kit was evaluated against the gold standard serological assay to validate its accuracy and reliability. All analyses were conducted at the UTH Virology Laboratory following the test kit's standard operating procedures, with laboratory quality control measures in place. For detailed assay specifications, refer to the Origene datasheet (https://cdn.origene.com/datasheet/ea100951.pdf).

In line with the ethical approval for this study, transport refunds were not provided to the study participants, as the target population consisted of children attending routine reviews at the hospital. However, transport refunds were given to healthcare workers involved in data collection. HIV-positive children were recruited from the UTH Pediatric HIV wards,

with their HIV status confirmed through existing medical records. HIV-negative children were recruited from the general pediatric ward, where their HIV-negative status was also verified through clear medical records. Data collection in the HIV wards was conducted by healthcare workers specializing in HIV care, ensuring accurate and ethical recruitment processes.

## Data analysis

The dataset was cleaned and included variables such as Body Mass Index (BMI), HIV status, Place of residence (Lusaka or outside Lusaka), Education status, Age, Sex and immunity status, vaccination timings (children who received their measles one at 9 months and measles two at 18 months were coded as on time, otherwise they were coded as not on time, coinfections (Children who presented infections such as malaria, diarrhea and tuberculosis) and place of residence with a total of 200 participants. Descriptive statistics were performed to observe the characteristics of variables (numbers and percentages were reported as the variables were categorized. Since the condition of the chi-square were met, associations between categorical variables were evaluated using the chi-squared test of independence. Univariate and multivariate logistic regression were employed to identify the variables that predicted waning immunity in the mining community. Using an investigator-led stepwise regression process, multiple logistic regression was utilized to identify the variables impacting waning immunity. To select the variables for the final multiple regression model, all of the predictor variables were first run through the multiple logistic regression command. Then, the predictor variables with the highest p-values were removed from the model one at a time until the predictor variables that best predicted the outcome remained in the model. Ultimately, the best-fit model was selected using the Bayesian information criteria (AIC and BIC) and Akaike's information criterion for the competing models. The model selected was the one with the lowest AIC and BIC values when compared to the other models. The crude odds ratio (cOR) and adjusted odds ratio (aOR) were displayed together with their respective 95% confidence intervals (CIs). A significance level of 0.05 was assigned to a p-value. Using R statistical analysis software version 4.4.1 data was analyzed.

## Ethical consideration

This study was approved by ERES Converge IRB (Ref No. 2024-Feb-025). Permission to conduct the study was granted by the National Health Research Authority (NHRA) (REF: NHREB 008/01/04/2024) and the Ministry of Health. In addition, written consent was obtained from the parents or guardians. In accordance with Zambian ethical guidelines, assent was obtained from children aged 7 years and above, as they are considered capable of understanding the study's purpose and procedures. For children below 7 years, including 2-year-olds, assent was not applicable, and only parental or guardian consent was obtained. All consent and assent forms were provided in both English and local languages (Bemba and Nyanja) to ensure understanding among participants and their guardians. The research assistant read and explained the benefits and possible risks of participating in the study to the participants. After gaining full knowledge, participants who agreed to take part were given a written consent form to sign. Respect for persons, beneficence, and justice were upheld throughout the study.

## Results

### Demographic characteristics

Table 1 summarizes the demographic and clinical characteristics of participants, stratified based on HIV status. Almost half of the sample (45%) were aged between 10 and 15 years. And majority (58%) were female and well nourished (75%). Majority (77.5%) were in school and had not received the vaccination on time (51%). Most children (76.0%) were breastfed and resided in Lusaka (93.0%). Waning immunity was significantly associated with age (p < 0.001), BMI (p < 0.001), education level (p < 0.001), breastfeeding status (p < 0.001), and province of residence (p = 0.013) among both HIV-negative and HIV-positive children. In contrast, no statistically significant association between the two groups was observed for vaccination timing (p = 0.12).

PLOS Global Public Health

**Table 1. Socio-demographic characteristics univariate analysis.**

| Demographic Characteristics | Overall (N) | HIV Negative (n (%)) | HIV Positive (n (%)) | P-Value |
|---|---|---|---|---|
| | N=200 | N=100 | N=100 | |
| **Age category** | | | | <0.001 |
| 2-4 years | 49 (24.5%) | 38 (38.0%) | 11 (11.0%) | |
| 5-9 years | 61 (30.5%) | 35 (35.0%) | 26 (26.0%) | |
| 10-15 years | 90 (45.0%) | 27 (27.0%) | 63 (63.0%) | |
| **Sex** | | | | 0.063 |
| Female | 116 (58.0%) | 65 (65.0%) | 51 (51.0%) | |
| Male | 84 (42.0%) | 35 (35.0%) | 49 (49.0%) | |
| **BMI Category** | | | | <0.001 |
| Malnutrition | 50 (25.0%) | 32 (32.0%) | 18 (12.0%) | |
| Well Nourished | 150 (75.0%) | 68 (68.0%) | 82 (82.0%) | |
| **Education Level** | | | | <0.001 |
| In school | 155 (77.5%) | 64 (64.0%) | 91 (91.0%) | |
| Not in school | 45 (22.5%) | 36 (36.0%) | 9 (9.00%) | |
| **Vaccination Timing** | | | | 0.12 |
| Not on time | 102 (51.0%) | 57 (57.0%) | 45 (45.0%) | |
| On time | 98 (49.0%) | 43 (43.0%) | 55 (55.0%) | |
| **Breastfed** | | | | <0.001 |
| No | 48 (24.0%) | 9 (9.00%) | 39 (39.0%) | |
| Yes | 152 (76.0%) | 91 (91.0%) | 61 (61.0%) | |
| *Coinfection | | | | 0.012 |
| Coinfected | 72 (36.0%) | 27 (27.0%) | 45 (45.0%) | |
| Not coinfected | 128 (64.0%) | 73 (73.0%) | 55 (55.0%) | |
| **Province of residence** | | | | 0.013 |
| Lusaka | 186 (93.0%) | 88 (88.0%) | 98 (98.0%) | |
| Other | 14 (7.00%) | 12 (12.0%) | 2 (2.00%) | |
| **HIV status** | | | | 0.001 |
| Retained Immunity | 100(50%) | 78(78%) | 38(38%) | |
| Waned Immunity | 100 (50%) | 22(22%) | 62(62%) | |

*In this study, co-infection refers to children who are presented with additional conditions such as tuberculosis, malaria, and diarrhea. Children with two or more infections were categorized as co-infected; otherwise, they were classified as not co-infected.

*Significant level was set at 95%

**Unadjusted and adjusted multivariate regression analysis**

Table 2 below shows the results for unadjusted and adjusted multivariate analysis. The adjusted analysis indicates that children aged 10–15 years had lower odds of retaining immunity compared to those aged 2–4 years (AOR= *0.228; 95% CI [0.076, 0.649], p < 0.006).* This means that children in this age group were more likely to lose their immunity compared to the younger age group. For children aged 5–9 years, the AOR was *0.854 (95% CI [0.300, 2.360], p = 0.762),* indicating no significant difference in the odds of waning immunity compared to children aged 2–4 years. Regarding HIV status, children who were HIV-positive had lower odds of retaining their immunity compared to their HIV-negative counterparts (AOR= *0.251; (95% CI [0.102, 0.596], p < 0.002).* This suggests that HIV-positive children were more likely to lose their immunity. Concerning breastfeeding, children who were breastfed had lower odds of waning immunity compared to those who were not breastfed (AOR= *0.356; 95% CI [0.140, 0.873], p = 0.0263).*

**Table 2. Unadjusted and adjusted multivariate regression analysis.**

| Factors (Reference) | Unadjusted OR | 95% CI | P - value | Adjusted OR | 95%CI | P-Value |
|---|---|---|---|---|---|---|
| Age Category | | | | | | |
| 2-4 years | Ref [1] | | | Ref [1] | | |
| 5-9 years | 1.105 | 0.502, 2.428 | 0.801 | 0.854 | 0.300, 2.360 | 0.762 |
| 10-15 years | 0.262 | 0.123,0.539 | <0.003 | 0.228 | 0.076, 0.649 | <0.006 |
| Sex | | | | | | |
| Females | Ref [1] | | | Ref [1] | | |
| Male | 1.324 | 0.755, 2.333 | 0.327 | 1.786 | 0.946, 3.830 | 0.075 |
| BMI Category | | | | | | |
| Malnourished | Ref [1] | | | Ref [1] | | |
| Well Nourished | 0.454 | 0.230,0.871 | 0.019* | 0.90 | 0.838,1.080 | 0.474 |
| Breastfeeding | | | | | | |
| No | Ref [1] | | | Ref [1] | | |
| Yes | 0.626 | 0.322,1.201 | 0.250 | 0.356 | 0.140, 0.873 | 0.0263 |
| Education Status | | | | | | |
| In school | Ref [1] | | | Ref [1] | | |
| Not in school | 1.729 | 0.886, 3.439 | 0.112 | 0.714 | 0.245, 2.03 | 0.529 |
| Vaccination Timing | | | | | | |
| Not on time | Ref [1] | | | Ref [1] | | |
| On-time | 0.818 | 0.468,1.424 | 0.478 | 1.050 | 0.440, 2.560 | 0.916 |
| HIV Status | | | | | | |
| Negative | Ref [1] | | | Ref [1] | | |
| Positive | 0.360 | 0.201,0.634 | <0.001 | 0.251 | 0.102, 0.596 | <0.002 |
| Coinfections | | | | | | |
| Coinfected | Ref [1] | | | Ref [1] | | |
| Not coinfected | 0.808 | 0.497,1.584 | 0.688 | 0.551 | 1.298, 3.127 | 0.111 |
| Educational Status Caregiver | | | | | | |
| Never been to school | Ref [1] | | | Ref [1] | | |
| Primary | 0.421 | 0.153,1.086 | 0.080 | 0.450 | 0.120, 1.570 | 0.220 |
| Secondary | 0.450 | 0.160,1.192 | 0.116 | 0.438 | 0.108, 1.670 | 0.234 |
| Tertiary | 0.480 | 0.167,1.300 | 0.157 | 0.615 | 0.141, 2.570 | 0.509 |
| Province of Residence | | | | | | |
| Within Lusaka | Ref [1] | | | Ref [1] | | |
| Outside Lusaka | 0.543 | 0.161,1.635 | 0.290 | 0.432 | 0.089, 201 | 0.285 |

## Factors associated with measles immunogenicity

Table 3 shows the results of the multivariable analysis for factors associated with measles immunogenicity. We adjusted for the following confounders using machine-led regression analysis: BMI Category, Education Status, Vaccination Timing, Coinfections, and Caregiver Educational Status. These variables were included in the model and adjusted for during the stepwise regression process. However, they were removed from the final model due to high p-values, indicating a lack of significant contribution to the outcome variable. The final model showed that age, HIV status, and history of breastfeeding were significantly associated with measles immunogenicity. Children aged 10–15 years had significantly lower odds of losing retaining immunity compared to those aged 2–4 years (AOR = 0.270; 95% CI [0.114, 0.618], p = <0.002). Regarding HIV status, children living with HIV infection had significantly lower odds of retaining immunity compared to those who are HIV-negative (AOR=0.290 (95% CI [0.137, 0.594], p < 0.001). Children who were breastfed had lower odds of waning

**Table 3. Multivariable analysis.**

| Factors (Reference) | OR | 95%CI | P-Value |
|---|---|---|---|
| Age Category | | | |
| 2-4 years | Ref [1] | | |
| 5-9 years | 1.15 | 0.493, 2.686 | 0.742 |
| 10-15 years | 0.270 | 0.114, 0.618 | <0.002 |
| Sex | | | |
| Females | Ref [1] | | |
| Male | 1.798 | 0.944, 3.499 | 0.077 |
| Breastfeeding | | | |
| No | Ref [1] | | |
| Yes | 0.336 | 0.147, 0.738 | 0.007 |
| HIV Status | | | |
| Negative | Ref [1] | | |
| Positive | 0.290 | 0.137, 0.594 | <0.001 |
| Province of Residence | | | |
| Within Lusaka | Ref [1] | | |
| Outside Lusaka | 0.250 | 0.066, 0.859 | 0.031 |

immunity compared to those who were not breastfed, with an (AOR=*0.336 (95% CI [0.147, 0.738], p = 0.007)*. Majority of the study participants were Lusaka residents limiting these finding to Lusaka and not other settings.

Sensitivity analyses were conducted to evaluate the effect of varying age categorizations (e.g., 2–4, 5–7, 8–11, 12–15 years). Children aged 12–15 years had significantly lower odds of retaining immunity compared to younger children (AOR: 0.159; 95% CI: 1.049-1.631, p=0.002). Sensitivity analyses confirmed these trends across different age categorizations.

## Discussion

This study examined the factors affecting measles vaccine immunogenicity in children aged 2–15 years at the University Teaching Hospital in Lusaka, Zambia. HIV-negative children had higher odds of retaining immunity compared to HIV-positive children. Similarly, children who were breastfed were more likely to retain immunity. This study highlights the importance of breastfeeding and age-appropriate vaccination strategies in enhancing immunity retention.

In addition, Our findings show that age significantly affects the immune response to measles vaccination. Older children aged 10–15 years were significantly more likely to lose their immunity compared to younger age groups, particularly those aged 2–9 years. The sensitivity analysis using alternative age groupings (2–4, 5–7, 8–11, and 12–15 years) confirmed the robustness of our findings. While no significant differences in immunity retention were observed for the 5–7 and 8–11 years groups compared to 2–4 years, children aged 10–15 years exhibited significantly lower odds of retaining immunity (OR: 0.159, p = 0.002). This suggests that waning immunity is most pronounced in late childhood and early adolescence.

The finding that immunity reduction is not uniform across intermediate age groups may be influenced by several factors. Biologically, immune memory dynamics and maturation may differ across these groups, with some age groups experiencing slower immunity decline due to recent vaccination or natural boosting from subclinical exposure. Differences in exposure to pathogens, nutritional status, or co-infections may further contribute to the observed variability. Additionally, smaller sample sizes in intermediate age categories might have limited statistical power to detect significant trends. These factors highlight the need for further studies to explore the interplay between biological, environmental, and healthcare-related factors influencing immunity retention.

This finding is consistent with various studies which have indicated that younger children, especially those below 5 years of age, show stronger immune responses to measles vaccination due to maternal antibodies by age two [19,20,21],

On the other hand, older children (10 years and above) may experience waning measles immunity over time [22,23].This finding was consistent with our study's finding that several older children had lower odds of retaining measles immunity after vaccination. In addition, literature has also shown that children vaccinated at younger ages may be at risk of waning immunity as they grow older [18,24]. These findings may call for further investigations, especially in children who receive booster doses as early as six months.

Another key finding for this study was HIV status and how it affected measles immunity among the HIV infected children. Literature has shown that even with viral suppression, HIV infected children [20,25] are more vulnerable to severe measles infections [26,27]. Over time, HIV-infected children show diminished vaccine-induced immunity compared to their uninfected counterparts, requiring closer monitoring [28,29]. These findings are consistent with previous studies which reported similar findings [30]. The finding highlights the importance of age-appropriate vaccination schedules in maintaining measles immunity over time especially in HIV infected children. Introducing booster doses for HIV infected children at age 10–15 would help contribute to strengthened immunity. In Zambia, the routine immunization program aims to reach 95% of children annually with two doses of measles-containing vaccine (MCV) administered at 9 and 18 months [14]. However, for HIV-infected children, additional booster doses or earlier vaccination at 6 months may be necessary to enhance protection and address the potential for waning immunity as they age [29]. This is particularly critical in regions with high HIV prevalence, where maternal antibodies may wane earlier, rendering children more susceptible to measles infection [30]. Previous studies have shown a strong retention of immunity in older children after receiving booster vaccine as a protective measure [28].

Breastfeeding was associated with a higher likelihood of retained immunity against measles, aligning with previous studies that suggest a protective role of breastfeeding in immune health. In this study, breastfeeding data were limited to a binary measure (breastfed or not), and details on breastfeeding duration or exclusivity were not included. However, as observed in other research, these initial findings hint at the potential contribution of passive immunity and the immunological benefits of breastfeeding in strengthening immune responses to vaccinations. This study contributes to the growing evidence supporting breastfeeding as a beneficial factor in immunity retention, although further research is needed to clarify its mechanisms and determine optimal breastfeeding practices that enhance vaccine efficacy. This finding aligns with the theory that nutritional status can influence immune response and that malnutrition impairs immunity [31]. On the contrary, a study indicated that maternal antibodies may inhibit the infant's immunological response to the measles vaccination, which could hinder the development of a strong antibody response in the child. Before the infant's immune system has a chance to properly react, maternal antibodies acquired through breastfeeding may destroy the live virus used in the measles vaccine [20]. Another study suggests that even if breastfeeding momentarily disrupts certain vaccine reactions, some research contends that it may strengthen an infant's immune response by exposing them to additional immunomodulatory elements. Thus, breastfeeding should not be discouraged; nevertheless, for maximum immunity, immunization scheduling may need to be carefully considered [32].

Lastly, most of the study participants resided in Lusaka Province, which limits the findings to urban settings and underscores the need for similar studies in rural areas to provide a more comprehensive understanding of geographic variability in measles immunity."

Several limitations associated with this study are worth considering. First, the study did not assess the quality of vaccine administration, including adherence to cold chain protocols and proper vaccine handling, which may have affected vaccine efficacy and contributed to waning immunity. Vaccines are expected to be stored between +2 and +8 °C, but exposure to sub-zero temperatures can damage heat-sensitive vaccines like Measles, Tetanus, DTP, and Hepatitis B [33–35]. Maintaining cold chain is essential to ensuring vaccine quality and effectiveness. However, vaccine exposure to suboptimal temperatures is a widespread challenge affecting vaccine efficacy in low-resource settings such as Zambia [36,37].

The study was conducted in Lusaka district which is predominantly urban, thereby limiting the generalizability of the findings to the wider Zambian population, especially in rural regions where access to healthcare and vaccination practices may vary considerably. Second, the lack of genomic sequencing of measles IgG samples limited the understanding of specific viral strains and mutations that may contribute to waning immunity. Mitigating these limitations in forthcoming research could augment our comprehension of measles immunity and facilitate the formulation of more precise public health interventions.

Despite these limitations, this study has various strengths that substantially enhance the comprehension of measles vaccine immunogenicity in children. First, the broad age range of sampled children facilitated the identification of age-related patterns in diminishing immunity, uncovering essential insights into the variations of immunity throughout different developmental stages. Second, the emphasis on HIV status as a critical factor yielded significant insights into the heightened susceptibility of HIV-positive children, underscoring the necessity for tailored vaccination programs and supplementary booster doses for this population. Third, the study also encompassed factors like breastfeeding and nutritional status, providing a more thorough understanding of broader health determinants and their impact on immune responses. The incorporation of province of residence as a variable underscored potential regional disparity, which is essential for formulating region-specific public health strategies. The alignment of the findings with existing literature supports the credibility of the results, while evidence-based recommendations, including booster vaccinations for HIV-infected children, offer practical guidance for enhancing measles control.

## Conclusion

These findings indicate that age, breastfeeding, and HIV status significantly influence the retention of measles immunity in children aged 10–15 years. Younger children and those without HIV exhibited higher odds of maintaining immunity. Additionally, breastfeeding demonstrated a potential protective role in enhancing immunity retention. Selected sub-districts in Lusaka were associated with waning immunity, highlighting the need for further investigation with a more diverse population sample. These results emphasize the importance of targeted immunization strategies, including booster doses for older and HIV-infected children in Zambia, to sustain immunity. Future research utilizing experimental designs is warranted to explore the underlying mechanisms contributing to immunity retention.

## Acknowledgments

Gratitude is extended to the staff and management of the University Teaching Hospital in Lusaka, the National Virology Laboratory, the ZNPHI Reference Lab, the Pediatric Centre of Excellence, and the University of Zambia Institute of Distance Education for their support and collaboration on this project.

## Author contributions

**Conceptualization:** Priscilla Nkonde Gardner, Jimmy Hangoma, Roma Chilengi, Muzala Kapina, Musole Chipoya.

**Methodology:** Cephas Sialubanje, Davie Simwaba.

**Supervision:** Jimmy Hangoma, Cephas Sialubanje, Isaac Fwemba.

**Writing – original draft:** Priscilla Nkonde Gardner, Jimmy Hangoma, Cephas Sialubanje.

**Writing – review & editing:** Cephas Sialubanje, Roma Chilengi, Muzala Kapina, Musole Chipoya, Kelvin Mwangilwa, Lillian Lamba, Musaku Mwenechanya, Rodgers Chilyabanyama, Mpanga Kasonde, Soo Young.

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
