## [Editor Report · Decision Letter 0]

1 Nov 2024

PGPH-D-24-02312

Factors Associated with Measles Vaccine Immunogenicity in Children at University Teaching Hospitals, Lusaka, Zambia

Dear Dr. Nkonde Gardner,

Thank you for submitting your manuscript to PLOS Global Public Health. After careful consideration, we feel that it has merit but does not fully meet PLOS Global Public Health’s publication criteria as it currently stands. Therefore, we invite you to submit a revised version of the manuscript that addresses the points raised during the review process. Please ensure that your decision is justified on PLOS Global Public Health’s publication criteria and not, for example, on novelty or perceived impact. ========================================================================Editor Comments:

Before I send this out for peer review some comments:

Could you make the abstract methods more in line with your specific research question?

For tables - can you specify co-infections further?

Did you collect any info on measles disease status? Like past history of measles disease.

Do you have any further comments or questions about how breastfeeding would be detrimental to measles antibody status

---

## [Decision Letter · Decision Letter 1]

3 Jan 2025

PGPH-D-24-02312R1

Factors Associated with Measles Vaccine Immunogenicity in Children at University Teaching Hospitals, Lusaka, Zambia

Dear Dr. Nkonde Gardner,

Thank you for submitting your manuscript to PLOS Global Public Health. After careful consideration, we feel that it has merit but does not fully meet PLOS Global Public Health’s publication criteria as it currently stands. Therefore, we invite you to submit a revised version of the manuscript that addresses the points raised during the review process.

First off - you can ignore reviewers 1 and 2 - that was my fault in how I sent out the manuscript to them.

I will note there was confusion among the reviewers in your manuscript because you had multiple versions with different titles. In your response, please only submit one version (and by that, I mean one version with track changes, and one cleaned).

Reviewers 4 and 6 have included attachments of their revisions.

We look forward to receiving your revised manuscript.

Kind regards,

Abram L. Wagner, PhD, MPH

Academic Editor

Journal Requirements:

Additional Editor Comments (if provided):

Reviewers' comments:

Reviewer's Responses to Questions

**Comments to the Author**

1. If the authors have adequately addressed your comments raised in a previous round of review and you feel that this manuscript is now acceptable for publication, you may indicate that here to bypass the “Comments to the Author” section, enter your conflict of interest statement in the “Confidential to Editor” section, and submit your "Accept" recommendation.

Reviewer #1: All comments have been addressed

Reviewer #2: All comments have been addressed

Reviewer #3: (No Response)

Reviewer #4: (No Response)

Reviewer #5: (No Response)

Reviewer #6: (No Response)

Reviewer #7: (No Response)

2. Does this manuscript meet PLOS Global Public Health’s publication criteria ? Is the manuscript technically sound, and do the data support the conclusions? The manuscript must describe methodologically and ethically rigorous research with conclusions that are appropriately drawn based on the data presented.

Reviewer #1: Yes

Reviewer #2: Yes

Reviewer #3: Yes

Reviewer #4: Partly

Reviewer #5: Yes

Reviewer #6: Yes

Reviewer #7: Partly

3. Has the statistical analysis been performed appropriately and rigorously?

Reviewer #1: Yes

Reviewer #2: Yes

Reviewer #3: Yes

Reviewer #4: Yes

Reviewer #5: Yes

Reviewer #6: Yes

Reviewer #7: No

4. Have the authors made all data underlying the findings in their manuscript fully available (please refer to the Data Availability Statement at the start of the manuscript PDF file)?

Reviewer #1: Yes

Reviewer #2: Yes

Reviewer #3: Yes

Reviewer #4: Yes

Reviewer #5: No

Reviewer #6: Yes

Reviewer #7: Yes

5. Is the manuscript presented in an intelligible fashion and written in standard English?

Reviewer #1: Yes

Reviewer #2: Yes

Reviewer #3: Yes

Reviewer #4: Yes

Reviewer #5: Yes

Reviewer #6: Yes

Reviewer #7: Yes

6. Review Comments to the Author

Reviewer #1: The changes have been made clearly and properly.

The manuscript has some issues related with the analysis and these are made clearly visible.

Reviewer #2: all comments have been adresssed

Reviewer #3: Thank you for giving me the opportunity to review this paper. It is well written, well structured and will contribute to the body of knowledge on this topic.

1. Methods

a) Inclusion and exclusion criteria: For the history of measles infection in the past 6 months; this seems like an important inclusion/exclusion criteria. However, it’s only been mentioned in the abstract. I propose that this important criterion is added to/described in the methods section.

b) Algorithm Figure: I did not have access to the algorithm on the Preflight Analysis and Conversion Engine digital tool (there was no link to the “S1 Fig: Participant Algorithm”). Did this also include a breakdown of number of participants that were screen failures and the reasons for the screen failures? It will be important to discuss this briefly in the methods section.

2. Results:

a) What was the rationale behind removing Tables 1,2 and 3 from the main body of the paper and moving them to supplementary material? These tables would be more interesting to look at when reading the results section than having to look for them in supplementary materials. Nevertheless, if this was an editorial decision, I would ask the authors to please revise the reference to these tables to Supplementary Tables 1, 2 and 3 in the results section of the main paper.

b) Factors associated with measles immunogenicity: “Children aged 10-15 years had significantly lower odds of losing retaining immunity compared to those aged 2-4 years”. Consider revising the text to “Children aged 10-15 years had significantly lower odds of retaining immunity compared to those aged 2-4 years”.

c) Children who were breastfed had lower odds of waning immunity compared to those who were not breastfed, with an (AOR=0.336 (95% CI [0.147, 0.738], p = 0.007). Consider removing the bracket before AOR.

d) Tables 2 and 3: Please add a footnote defining the asterix added to the p-values.

Discussion

a) “HIV-negative children had higher odds of retaining immunity compared to HIV positive children, and those who were breast underweight children also revealed higher odds of waning immunity”. Consider removing the word breast.

Reviewer #4: I am reviewing a draft dated from 17 December 2024, NOT the revision that was attached to this review software. This "newer" version was provided to me by PLOS editorial staff (Abram Wagner and Shaira Petancio). The title of the draft that I am reviewing is, "Predictors of Measles Vaccines Immunogenicity in Children at University Teaching Hospitals, Lusaka, Zambia." I have attached this version to this reply.

This is my first reading of the manuscript, so I cannot comment on whether the authors have responded. It should be noted that in communications with the editorial staff at PLOS, there may be multiple versions of this manuscript that are circulating among reviewers. In addition, this latest draft does not have Line Numbering, page numbers, nor are any of the Figures or Tables provided. Thus, my suggested revisions will attempt to locate where revisions need to be made. I am unable to comment on the Figure and Tables.

This article examines possible predictors/factors for waning immunity to Measles vaccines in Africa. The approach, methodologies, and analysis are sound. The conclusions and recommendations seem justified. I think the challenge will be to consolidate the various drafts and multiple inputs from reviewers.

SPECIFC COMMENTS: (I have hand-numbered the pages starting with the Title page, then the Abstract, Introduction, etc.)

ABSTRACT: This version does not include the subtitles (e.g. Background, Methods, etc). I assume that this is acceptable to PLOS.

MAIN TEXT (PAGE 3): In the Introduction, second paragraph, I would suggest revising the sentence to, "Several factors have been shown to affect the efficacy of measles immunizations such as THE QUALITY of vaccine storage and the cold chain."

PAGE 4. Rewrite the sentence, "Disparities in immunity levels between HIV-positive and HIV-negative children and adults in Zambia were noted in a seroprevalence STUDY, highlighting the importance......"

PAGE 5. Rewrite the sentence under the title, "Inclusion and Exclusion Criteria" as follows: "Children below the age of two years and above 15 years were excluded from the study, as were those with severe immunodeficiency other than HIV, such as primary immunodeficiency disorders. "

PAGE 6. Under the section Data Analysis, can the authors rewrite the first sentence? "Data was cleaned containing...." doesn't make sense.

PAGE 9. The last sentence before the Discussion needs to be rewritten. Possibly, "The majority of the study participants were residents of Lusaka which limits the findings."

PAGE 9 Under the DISCUSSION section in the first paragraph, the last sentence needs to be rewritten. It's not clear what "..breast underweight children...." means.

Reviewer #5: This study explores important factors influencing measles vaccine immunogenicity among children aged 2–15 years in Zambia, where a cross-sectional design was employed to evaluate the impact of age, HIV status, breastfeeding, and geographical residence. The findings are consistent with existing literature and this study in particular contributes to this literature by highlighting the potential value of tailored vaccination programs for HIV-positive children and the protective role of breastfeeding in immune health.

Overall, the manuscript is well-written and easy to read and follow. The descriptive statistics are well summarized into tables and graphs. Below are some concerns and suggestions for the authors to consider.

1. The analysis was overall well conducted and addressed the questions of interest. The only big question I have is regarding how different age categorizations may affect the results. I would expect immunity waning to happen continuously and somewhat consistently over time after certain point post vaccination, so it is surprising to see no immunity reduction at all in the 5-9 years group. One way to better understand the pattern behind it would be to run a couple of sensitivity analyses that look into different ways of binning the age groups, e.g., 2-4, 5-7, 8-11, 12-15.

2. The study mentioned controlling for potential confounders, but it was not clear which confounders did the authors control for. Were they the same variables as listed in Table 2 or were there any additional confounders?

3. The authors suggested tailored vaccination programs and supplementary booster doses based on the study findings. While I am personally positive about the potential of these programs, I would love to see the authors providing more quantitative evidence using data from this study. For example, if we were binning age groups in a different way as mentioned above, can we possibly identify the best age group to roll out a booster dose?

Reviewer #6: Title:

Factors Associated with Measles Vaccine Immunogenicity in Children at University Teaching Hospitals, Lusaka, Zambia.

General comments

1. Did collect assents from all the children included in the study? From what age the assent is valid in Zambia? Can a child of 2 years old provide a valid assent?

2. Please check the whole document and insert the references properly. It is not appropriate to insert reference after a dot (.) and neither inserting multiple references separately [(2), (3). Below is an example: “Immunization remains the most effective strategy for reducing morbidity and mortality from infectious diseases worldwide. However, the immune response to vaccines can vary considerably due to factors such as age, HIV status, and socio-demographic conditions. (2) (3).” This should be inserted like this: “… HIV status, and socio-demographic conditions (2, 3).

3. Check the tense used in writing in the whole document. There is a tension and mix of present and past tenses.

Abstract

“Residing in Lusaka was associated with lower immunity retention compared to children outside the province; …” => I am not sure this sentence is properly structured. Maybe say people who live in Lusaka showed lower immunity retention than those living outside …”.

Introduction

“The measles prevention initiative in Zambia seeks to reach 95 percent of children each year

through the routine immunization program”. => it is good to provide reference here.

The introduction is dominated by information about HIV positive and HIV negative children. It could better to have also information from the literature about breastfeeding, age in relation to measles immunity.

It would be better to have the aim of the study ang research questions at the end of the introduction.

Methods

“This comparative analytical cross-sectional study design was conducted from the 2nd of April to the 31st of July 2024”. => Can author write this section about study design properly. One sentence is not sufficient and it is not clear. Is it a retrospective study or prospective?

“The hospital recorded over 600 measles cases with about 20 deaths between 2022 and 2023

(19) ’’ => Can this sentence be part of the introduction (in the background) and explain this situation very well as part of the rationale for this study?

Data collected from April to July 2024, was this done prospectively? Was there a time and transport compensation for participants?

For inclusion and exclusion criteria, what about children who did not know their HIV status at the time of data collection? Were they tested by authors or just excluded from the study? How did authors approach participants for their enrollment in the study? Was there a community engagement programme?

“Children below the age of two years and above 15 years were and those with severe

immunodeficiency other than HIV, such as primary immunodeficiency disorders were excluded

from the study“. => what indicator did authors use for this?

“Univariable and multivariable logistic regression were employed to identify the

variables that predicted waning immunity in the mining community”. => I am not sure what this sentence means. Do you want to say “Univariate and multivariate analysis” ?

Results

It would be better to have a brief description of the results after the table.

Reviewer #7: Include a statement on the justification of the study in the introductory section.

The terms immunity and immunogenicity are used interchangeably. Is this accurate?

Re-write to exclude repetitions such as age, and “Lusaka in Lusaka”

How was measles immunity measured?

What was the definition of “retained immunity”

What was the basis for categorizing participants as having “retained” or “waning” measles immunity?

Introduction

What is the global burden of measles (incidence, disability, mortality)?

What is the key theme in the first paragraph? Present one theme per paragraph

“Research has demonstrated that the efficacy of the measles vaccine can diminish with age” The reference (6) by Franconeri et al studied vaccine effectiveness and not efficacy as adduced by the authors. For clarity, the authors may need to indicate that measles seropositivity is used as a marker of vaccine effectiveness

Please revisit references 16 and 17 used to the statement, “Vaccines …. Exposure to sub-zero temperatures can damage heat-sensitive vaccines…” Also, is it accurate that exposure to sub-zero temperatures (or rather warmer temperatures) damages heat sensitive vaccines?

Check the reference manager, where are references 9 – 15? These don’t appear anywhere in the document before references 16 and 17 above.

How many children are living with HIV in Zambia? This should help us appreciate the seriousness of HIV/measles co-infection.

Are there challenges in measles vaccine coverage among children living with HIV? Is vaccine coverage different between HIV positive and negative individuals?

Methods

Study setting – please present the measles vaccine coverage, measles incidence and HIV prevalence of the catchment areas of these hospitals.

Study population and sampling procedure

What does the following statement mean? “The measles prevalence rate for the infected group ( 1) was 47.2% (0.472)(20) while the uninfected group ( 2) had a prevalence rate of 85% (0.85) (21)” Who are the infected group? Does infected mean, infected with HIV?

Inclusion and exclusion criteria

How was HIV negative status confirmed for HIV un-infected children?

The following statement is ambiguous, “Children below the age of two years and above 15 years were and those with severe immunodeficiency other than HIV…..”

What does the following statement mean, “There was significant difference between the HIV-negative and HIV-positive children about age (p<0.001),….”?

What do the authors mean by, “… retaining immunity…”?

How was measles immunogenicity measured?

7. PLOS authors have the option to publish the peer review history of their article (what does this mean? ). If published, this will include your full peer review and any attached files.

**Do you want your identity to be public for this peer review?** For information about this choice, including consent withdrawal, please see our Privacy Policy .

Reviewer #1: **Yes: ** Dr. Md. Abdullah yusuf

Reviewer #2: **Yes: ** ESTER LILIAN ACEN

Reviewer #3: **Yes: ** Grace Mambula

Reviewer #4: **Yes: ** Paul R De Lay, MD, DTM&H (Lond)

Reviewer #5: No

Reviewer #6: No

Reviewer #7: No

---

## [Decision Letter · Decision Letter 2]

11 Feb 2025

PGPH-D-24-02312R2

Predictors of Measles Vaccine Immunogenicity in Vaccinated Children at University Teaching Hospitals, Lusaka, Zambia

Dear Dr. Nkonde Gardner,

Thank you for submitting your manuscript to PLOS Global Public Health. After careful consideration, we feel that it has merit but does not fully meet PLOS Global Public Health’s publication criteria as it currently stands. Therefore, we invite you to submit a revised version of the manuscript that addresses the points raised during the review process.

Two of the reviewers request some minor revisions - please see their comments below.

Could you please revise the manuscript to carefully address the concerns raised?

We look forward to receiving your revised manuscript.

Kind regards,

Steve Zimmerman, PhD

PLOS Staff Editor

Journal Requirements:

Additional Editor Comments (if provided):

Reviewers' comments:

Reviewer's Responses to Questions

**Comments to the Author**

1. If the authors have adequately addressed your comments raised in a previous round of review and you feel that this manuscript is now acceptable for publication, you may indicate that here to bypass the “Comments to the Author” section, enter your conflict of interest statement in the “Confidential to Editor” section, and submit your "Accept" recommendation.

Reviewer #1: All comments have been addressed

Reviewer #2: All comments have been addressed

Reviewer #3: All comments have been addressed

Reviewer #5: All comments have been addressed

Reviewer #6: (No Response)

2. Does this manuscript meet PLOS Global Public Health’s publication criteria ? Is the manuscript technically sound, and do the data support the conclusions? The manuscript must describe methodologically and ethically rigorous research with conclusions that are appropriately drawn based on the data presented.

Reviewer #1: Yes

Reviewer #2: Yes

Reviewer #3: Yes

Reviewer #5: Yes

Reviewer #6: Yes

3. Has the statistical analysis been performed appropriately and rigorously?

Reviewer #1: Yes

Reviewer #2: Yes

Reviewer #3: Yes

Reviewer #5: Yes

Reviewer #6: (No Response)

4. Have the authors made all data underlying the findings in their manuscript fully available (please refer to the Data Availability Statement at the start of the manuscript PDF file)?

Reviewer #1: Yes

Reviewer #2: Yes

Reviewer #3: Yes

Reviewer #5: Yes

Reviewer #6: Yes

5. Is the manuscript presented in an intelligible fashion and written in standard English?

Reviewer #1: Yes

Reviewer #2: Yes

Reviewer #3: Yes

Reviewer #5: Yes

Reviewer #6: Yes

6. Review Comments to the Author

Reviewer #1: The corrections have been made properly. The conclusion should contain only the main findings in relation with the objective of the study. The methodological part have shown that this is performed properly. However, the ELISA kit should be evaluated with the gold standard. Otherwise this will hamper the outcomes of the study.

Reviewer #2: All comments have been adressed

Reviewer #3: (No Response)

Reviewer #5: All comments have been addressed - great work. Please make sure to check for typos. For example, right before the Discussion section, the authors wrote:

Sensitivity analyses were conducted to evaluate the effect of varying age categorizations (e.g., 2-4, 5-7, 8-11, 12-15 years). Children aged "10-15 years" had significantly lower odds of retaining immunity compared to younger children (AOR: 0.159; 95% CI: 1.049-1.631, p=0.002).

I think the start of the second sentence should read as "Children aged 12-15 years" in the sensitivity analysis. And the CI (both ends greater than 1) does not make sense here for an OR of 0.159.

Reviewer #6: Title:

Factors Associated with Measles Vaccine Immunogenicity in Children at University Teaching Hospitals, Lusaka, Zambia.

Abstract

OK

Introduction

“Measles also results in severe complications such as pneumonia, encephalitis, and subacute sclerosing panencephalitis (SSPE), a progressive and fatal brain disorder that can develop years after the initial infection”. => This sentence needs reference.

“While these studies measure vaccine effectiveness in real-world settings using seropositivity as a marker, the findings underscore the need for careful monitoring of immunity levels across age groups and consideration of booster doses to sustain population-level immunity (8,9) for example, a study by Puthanakit in Thailand found that older children had significantly lower rates of measles seropositivity compared to younger children, highlighting the need for age appropriate vaccination strategies (10)”. => This sentence is too long, and you can split it into 2.

For the third research question: “How do socio-demographic factors affect the retention of immunity in this population? “, is it possible to give some examples of socio-demographic factors? Do not assume that people know them. Aren’t age and breastfeeding part of these factors?

Methods

“To be included in the study, participants needed to be meet the following criteria: …” => correct grammar in this sentence.

“Data was collected from participants who visited the hospital between April and July using a complete enumeration”. => This is a repetition; it is already written above.

“Figure 1: Participant Algorithm …” => starting from this sentence, this paragraph should be part of inclusion and exclusion criteria or enrollment process. In the current version, it looks like being part of Ethics process.

Results

OK

Discussion

OK

Conclusion

Living in Lusaka was associated with waning immunity; however, … => put a dot after the word immunity and start However with a new sentence.

7. PLOS authors have the option to publish the peer review history of their article (what does this mean? ). If published, this will include your full peer review and any attached files.

**Do you want your identity to be public for this peer review?** For information about this choice, including consent withdrawal, please see our Privacy Policy .

Reviewer #1: **Yes: ** Dr. Md. Abdullah Yusuf

Reviewer #2: **Yes: ** ESTER LILIAN ACEN

Reviewer #3: **Yes: ** Grace Mambula

Reviewer #5: No

Reviewer #6: No

---

## [Decision Letter · Decision Letter 3]

10 Mar 2025

PGPH-D-24-02312R3

Predictors of Measles Vaccine Immunogenicity in Vaccinated Children at University Teaching Hospitals, Lusaka, Zambia

Dear Dr. Nkonde Gardner,

Thank you for submitting your manuscript to PLOS Global Public Health. After careful consideration, we feel that it has merit but does not fully meet PLOS Global Public Health’s publication criteria as it currently stands. Therefore, we invite you to submit a revised version of the manuscript that addresses the points raised during the review process.

Note that there are very minor comments from the editor and from reviewer 6.

We look forward to receiving your revised manuscript.

Kind regards,

Abram L. Wagner, PhD, MPH

Academic Editor

Journal Requirements:

Additional Editor Comments (if provided):

I have some minor edits for you. I have already looked through the peer review comments, and you seem to have adequately addressed them.

There are just a few structural issues I would want for you to address prior to me accepting this.

The abstract contains information it does not need to. The sentence " To enhance model parsimony and

interpretability, variables like carer education level, co-infections, and child education

status were excluded based on investigator-led decisions. These adjustments ensured

the analysis focused on the most relevant predictors of measles immunity retention

while minimising potential bias."

is unnecessary in the abstract, as is the explanation of the alpha level.

Instead I would have one or more sentences in the abstract explaining some of your descriptive results (like number of children with and without HIV, and what their measles seropositivity was)

In the introduction - you have a paragraph on vaccine storage and cold chain. That is potentially interesting as a mechanism to explain lower measles vaccine immunogenicity among all - but not necessarily why it would be different in HIV+ vs HIV- individuals - so I would move this to the discussion.

I know this study has already gone through several rounds of review, and I think if you address the points above, I will not need to send it out for further peer review.

Reviewers' comments:

Reviewer's Responses to Questions

**Comments to the Author**

1. If the authors have adequately addressed your comments raised in a previous round of review and you feel that this manuscript is now acceptable for publication, you may indicate that here to bypass the “Comments to the Author” section, enter your conflict of interest statement in the “Confidential to Editor” section, and submit your "Accept" recommendation.

Reviewer #5: All comments have been addressed

Reviewer #6: (No Response)

2. Does this manuscript meet PLOS Global Public Health’s publication criteria ? Is the manuscript technically sound, and do the data support the conclusions? The manuscript must describe methodologically and ethically rigorous research with conclusions that are appropriately drawn based on the data presented.

Reviewer #5: Yes

Reviewer #6: Yes

3. Has the statistical analysis been performed appropriately and rigorously?

Reviewer #5: Yes

Reviewer #6: Yes

4. Have the authors made all data underlying the findings in their manuscript fully available (please refer to the Data Availability Statement at the start of the manuscript PDF file)?

Reviewer #5: Yes

Reviewer #6: Yes

5. Is the manuscript presented in an intelligible fashion and written in standard English?

Reviewer #5: Yes

Reviewer #6: Yes

6. Review Comments to the Author

Reviewer #5: All my comments have been addressed nicely. Great work!

Reviewer #6: Title:

Factors Associated with Measles Vaccine Immunogenicity in Children at University Teaching Hospitals, Lusaka, Zambia.

Abstract

OK

Introduction

The sentence “Insight into these factors were important inform policy and interventions on measles vaccination programs in Zambia” should read “Insight into these factors were important to inform policy and interventions on measles vaccination programs in Zambia”.

Methods

The sentence “Participants who were included in this study meet the following criteria:…” should read “Participants who were included in this study met the following criteria:…”

Results

OK

Discussion

OK

Conclusion

OK

7. PLOS authors have the option to publish the peer review history of their article (what does this mean? ). If published, this will include your full peer review and any attached files.

**Do you want your identity to be public for this peer review?** For information about this choice, including consent withdrawal, please see our Privacy Policy .

Reviewer #5: No

Reviewer #6: No

---

## [Editor Report · Decision Letter 4]

14 Mar 2025

Predictors of Measles Vaccine Immunogenicity in Vaccinated Children at University Teaching Hospitals, Lusaka, Zambia

PGPH-D-24-02312R4

Dear Ms Nkonde Gardner,

We are pleased to inform you that your manuscript 'Predictors of Measles Vaccine Immunogenicity in Vaccinated Children at University Teaching Hospitals, Lusaka, Zambia' has been provisionally accepted for publication in PLOS Global Public Health.

Best regards,

Abram L. Wagner, PhD, MPH

Academic Editor